# High production of furfural by flash pyrolysis of C6 sugars and lignocellulose by Pd-PdO/ZnSO$_4$ catalyst

Qiaoqiao Zhou[1], Jinxing Gu[1], Jingwei Wang[1], Anthony De Girolamo[1], Sasha Yang[1] & Lian Zhang [ORCID][1] ✉

Furfural (C$_5$H$_4$O$_2$) is an important platform chemical for the synthesis of next-generation bio-fuels. Herein, we report a novel and reusable heterogeneous catalyst, Pd-PdO/ZnSO$_4$ with 1.1 mol% palladium (Pd), for the production of furfural by flash pyrolysis of lignocelluloses at 400 °C. For both dry and wet C6 cellulose and its monomers, the furfural yields reach 74–82 mol%, relative to 96 mol% from C5 xylan and 23–33 wt% from sugarcane bagasse and corncob. The catalyst has a well-defined structure and bifunctional property, comprising a ZnSO$_4$ support for the dehydration and isomerization of glucose, and a local core-shell configuration for metallic Pd$^0$ encapsulated by an oxide (PdO) layer. The PdO layer is active for the Grob fragmentation of formaldehyde (HCHO) from glucose, which is subsequently in-situ steam reformed into syngas (i.e. H$_2$ and CO), whereas the Pd$^0$ core is active in promoting the last dehydration step for the formation of furfural.

Furfural (C$_5$H$_4$O$_2$) is an important platform chemical that is traditionally derived from the pentosans (i.e. C5 sugars) within lignocellulosic biomass[1]. Due to its unique chemical structure, furfural can be used as a starting substrate for the synthesis of a broad range of high-value chemicals, e.g. bio-plastics[2], food additives, and pharmaceuticals[3,4] and bio-fuels such as gasoline[5,6], promoting the development of a carbon-constrained future. It has a global market of USD $556.74 million (~$1000/ton) in 2022, with a projected compound annual growth rate (CAGR) of 7.0% in the forecast period of 2023–2030[7]. However, since the commercialization of the first technology, namely QuakerOats in 1921, furfural is only produced from the solvent-thermal method, i.e. solvolysis[8,9], where the batch-scale operation and/or a long residence time (e.g. ~1 h) induces a low efficiency (hence, a low throughput), the use of solvent and mineral acid (as the catalyst) causes the discharge of a large quantity of wastewater that is environmentally concerning, and more importantly, the C6 sugars within the feedstock mostly remain unreacted and are dumped as furfural residue waste that significantly negates the process cost-effectiveness and its environmental concerns.

From the chemical reaction perspective, the conversion of C5 sugars to furfural is simple, requiring only dehydration in both low-temperature solvolysis (e.g. 140–240 °C) and high-temperature pyrolysis (e.g. 400–700 °C)[8,10] in Fig. 1a, where an acidic catalyst such as mineral acid (e.g. sulphuric acid) is usually employed[9]. In contrast, the conversion of C6 sugars (e.g. glucose, C$_6$H$_{12}$O$_6$) to furfural is more complex, requiring both dehydration and Grob fragmentation for the removal of hydroxyls (·OH) and the side group formaldehyde (HCHO), respectively. So far, the C6 sugars were mostly converted to 5-hydroxymethyl furfural (5-HMF, C$_6$H$_6$O$_3$) by solvolysis[11–17], or to levoglucosan (LGA, C$_6$H$_{10}$O$_5$)[18–20], and/or levoglucosenone (LGO, C$_6$H$_6$O$_3$)[11,21] by pyrolysis in Fig. 1b, A maximum selectivity of 15.9% for furfural was only achieved from the flash pyrolysis of cellulose at 600 °C[16,22–24].

Herein, we report a novel heterogeneous catalyst, namely, Pd-PdO/ZnSO$_4$ for the doping of 1.1 mol% Pd, on zinc sulfate (ZnSO$_4$) to produce furfural from the flash pyrolysis of C6 sugars (Fig. 1c). Compared to batch-scale solvolysis, flash pyrolysis facilitates an improved process cost-benefit ratio via continuous operation, thereby achieving a large throughput and high efficiency and alleviating the environmental concerns associated with wastewater generated by mineral acids and used solvents[25]. The heterogeneous catalyst developed here is also expected to be bifunctional, selectively catalyzing both

[1]Department of Chemical & Biological Engineering, Monash University, Wellington Road, Clayton, VIC, Australia. ✉e-mail: lian.zhang@monash.edu

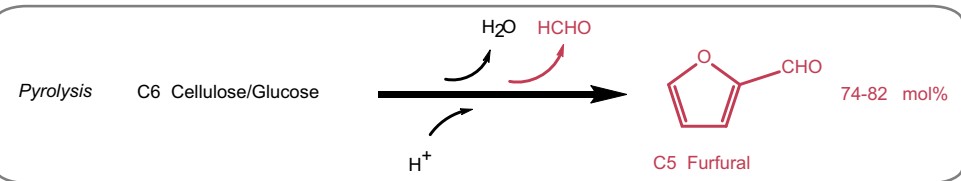

**Fig. 1 | Reaction pathways of different types of feedstocks.** Panel **a** for C5 sugars by solvolysis and pyrolysis. Panel **b** for C6 sugars by solvolysis (b-1) and pyrolysis (b-2); and Panel **c** for the conversion of C6 sugars by catalytic pyrolysis with the Pd-PdO/ZnSO$_4$ catalyst developed in this work.

dehydration and Grob fragmentation from the C6 sugar molecules. The ZnSO$_4$ support aims to dehydrate C6 molecules, as suggested by studies on C5 sugars, in which sulfates have proven to be the most effective[26–32], and the cation Zn$^{2+}$ (in the form of ZnCl$_2$)[30,33–37] has a higher dehydration activity than its counterparts, including ferric iron (Fe$^{3+}$)[30,38–43] and copper (Cu$^{2+}$)[28,38,41,44]. On the other hand, the Pd dopant is expected to selectively cleave the C-C bond from the side group formaldehyde, as has been suggested by the production of furan from furfural using a 2% Pd/carbon catalyst[45] and in petroleum cracking[46]. Moreover, once cleaved, the small carbon fragments are also reformed in situ by water molecules that are either present within the feedstock or derived from the dehydration reaction, as suggested by the Pd-catalyzed steam reforming of methanol[47] and methane[48]. Consequently, the catalyst is expected to have a high stability upon repeated usage. To the best of our knowledge, no studies have been conducted on the use of this specially designed catalyst.

A variety of advanced facilities were employed in this study, including a TGA-MS for temperature-programmed pyrolysis and real-time detection of any intermediates formed; a Pyroprobe micro-reactor coupled with GC-TCD/FID/MS detector (i.e. Py-GC-TCD/FID/MS) for the catalytic flash pyrolysis at an averaged heating rate of ~ 140 °C/s[49], and real-time detection and quantification of the liquid and permanent gas products; as well as a batch-scale, fixed-bed reactor with a capacity of grams per batch to validate the results from the two

micro-reactors. More specifically, a broad range of feedstocks from real biomass (corncob and sugarcane bagasse, both dry and wet) to pure compounds including cellulose, xylan, glucose and allose were tested to demonstrate and optimize the activity of our specially designed catalyst. To establish the reaction pathways, $^{13}$C-labeled glucose and the intermediates of C6 sugars including LGA, LGO and 5-HMF were also tested to trace the C-C cleavage position and the respective products. Finally, density functional theory (DFT) calculations, in combination with advanced characterization including XAS, XPS and TEM were employed to explore the active site and catalysis mechanism underpinning the activity of our specially designed catalyst and the reactions on its surface.

## Results and Discussion
### Properties of the Pd-PdO/ZnSO$_4$ catalyst

A series of catalysts were first prepared by doping Pd (0.4–3.4 mol%) on ZnSO$_4$, following the wet impregnation method and subsequent annealing at 550 °C in argon (Ar), as detailed in the Methods section. By bench-top XRD analysis, the (111), (200), (220) and (311) facets of metallic Pd$^0$ (JCPDS: 87-0639) were observed in the case of 3.4 mol% Pd doped on the ZnSO$_4$ support (Figs. S1a, b of Supplementary Information, SI), whereas those doped with 0.4–1.1 mol% Pd failed to show any Pd-bearing species. The detection of PdO by XRD is infeasible since its principal peaks (002) and (011) overlap with the (220) and (121) peaks

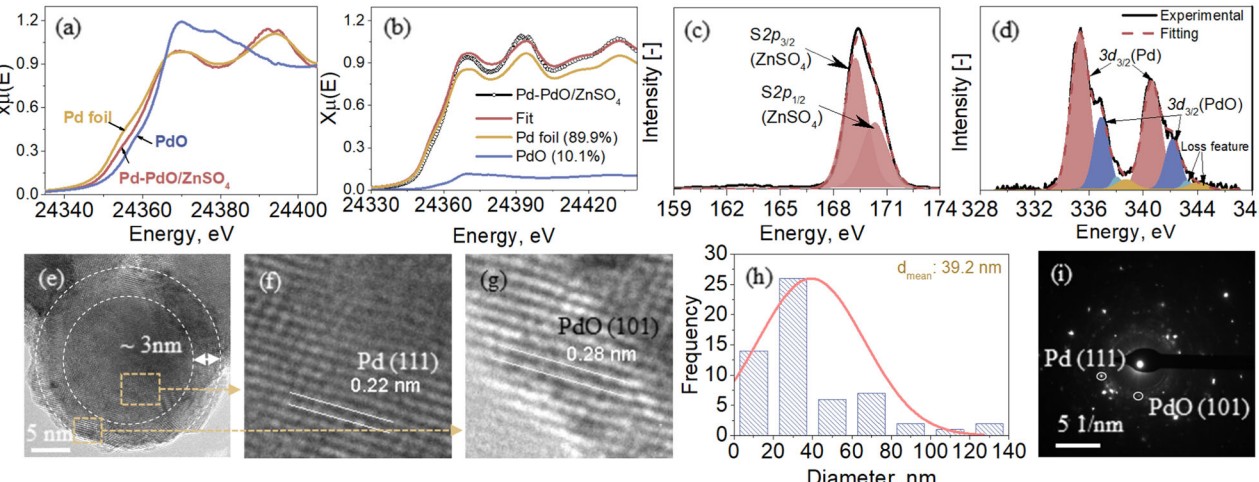

**Fig. 2 | Characteristics of 1.1 mol% Pd-doped ZnSO₄ catalyst.** Panel **a** for Pd K-edge XANES spectra of the catalyst and two standards - Pd foil and PdO. Panel **b** for the linear combination fitting (LCF) of the XANES spectra of the catalyst. Panels **c** and **d** for the XPS spectra of S$_{2p}$ and Pd$_{3d}$, respectively. Panels **e**–**g** for the high-magnification TEM observations of a typical Pd particle in the catalyst. Panel **h** for particle size distribution of Pd. Panel **i** for the SAED pattern of Pd clusters in the catalyst.

of ZnSO₄ at 33.64° and 34.00°, respectively. In addition, the temperature-programmed high-temperature XRD (HT-XRD, Fig. S2a) analyses confirmed the loss of chemical water from the ZnSO₄·H₂O precursor from 350 °C and the high stability of the resultant dehydrated ZnSO₄ up to 650 °C, as evidenced by its featured peaks at 10.60° (110), 12.42° (111), 16.56° (220), 16.74° (121) and 18.09° (301) against the ZnSO₄ standard (JCPDS: 96-900-9832).

A detailed characterization was conducted for the 1.1 mol% Pd-doped ZnSO₄ in Fig. 2. The Pd K-edge XANES analysis in Fig. 2a confirmed the coexistence of metallic Pd⁰ and PdO on the ZnSO₄ support, with molar percentages of 89.9% and 10.1% of total Pd, respectively (Fig. 2b, based on linear combination fitting using two standards, Pd foil and PdO). Fig. 2c for XPS S$_{2p}$ surface analysis confirmed the stability of the ZnSO₄ support after high-temperature annealing, whereas Fig. 2d for XPS analysis of Pd$_{3d}$ also confirmed the coexistence of metallic Pd⁰ and PdO on the surface, with abundances of 72.0% and 28.0%, respectively. Inferably, the difference between these values and the XAS results supports a surface enrichment of PdO. Additionally, the high-resolution TEM (HRTEM) images (Fig. 2e) showed a spherical shape for the Pd-bearing particles with an average particle diameter of ~40 nm (Fig. 2h) on the support surface. The metallic Pd⁰ with a d-space of 0.22 nm for its (111) facet was confirmed in Fig. 2f, whereas PdO with a featured d-space of 0.28 nm for its (101) facet was confirmed in Fig. 2g, with the SAED pattern shown in Fig. 2i[50]. Furthermore, as indicated in Fig. 2e, the two Pd-bearing species are spatially distributed as a core-shell structure, with metallic Pd⁰ as the core and PdO as the shell with a thickness of approximately 3 nm. This unique structure for the Pd distribution agrees with the XPS Pd$_{3d}$ analysis and is further confirmed by the STEM observation in Fig. 3. The low-magnification observations in (a)–(c) confirmed the core-shell structure for multiple particles, whereas the elemental mapping in (d) confirmed a nearly identical distribution of Zn, S and O supporting a stable ZnSO₄ relative to a slight mismatch between Pd and O for an abundance of O compared to Pd on the edge of particles. Such a phenomenon can be attributed to the strong interaction between Pd and the ZnSO₄ support, as metallic Pd⁰ was not found by annealing the Pd precursor Pd(NO₃)₂·2H₂O alone in nitrogen (Fig. S2b). Alternatively, only PdO with a similar particle size of 40 nm was observed by XRD (Fig. S3a), XPS (Fig. S3b) and TEM (Figs. S3c–g) analyses. Inferably, a portion of the lattice oxygen in PdO was scavenged by the unsaturated S and/or Zn on the ZnSO₄ surface during the heat treatment in Ar. This

should be due to the difference in the oxophilicity of the three elements, 0.5 of S, 0.2 of Zn and 0.0 of Pd. Indeed, the DFT calculation results shown later in Fig. 8i confirmed that ZnSO₄ has the strongest adsorption for oxygen from the hydroxyl group, with an adsorption energy of −0.43 eV on its surface, relative to −0.33 eV for PdO and −0.22 eV for Pd at 400 °C.

The physicochemical properties of the 1.1 mol% Pd-doped ZnSO₄, including acidity, BET surface area and porosity, can be found in Table 1. The respective NH₃-TPD, pyridine-FTIR spectra and N₂ adsorption/desorption isotherms are provided in Figs. S4, S5. The average pore size is similar across all the catalysts, 5.5–5.7 nm. The Pd-PdO/ZnSO₄ catalyst has the largest BET surface area of 4.49 m²/g, relative to 3.51 m²/g for ZnSO₄ and 2.02 m²/g for pure PdO. Nevertheless, both the total surface area and micropore area are very low, ruling out the importance of shape selectivity based on the pore structure. More interestingly, the combination of Pd and ZnSO₄ led to the largest total acid concentration and the highest Brønsted to Lewis acid (B/L) ratio. Compared to the pure PdO reference, the strongest acid site with a peak at 600–800 °C (NH₃-TPD) was also significantly suppressed. Consequently, this should be beneficial for quick desorption of the furfural product and a high turnover for the catalyst.

## Activity of Pd-PdO/ZnSO₄ catalyst on the conversion of C5 and C6 sugars

The extensive parametric screening was first conducted to optimize the pyrolysis temperature (300–500 °C), doping amount of Pd (0.4–3.4 mol%) and mass ratio of catalyst to biomass (up to 16) with allose as the feedstock in Pyro-probe reactor. The results are shown in Figs. S6, S7 of SI. Unless specified elsewhere, all the experiments were conducted on a batch scale with the feedstock and/or catalyst being pre-loaded into the reactor. In terms of furfural yield, Fig. S6a confirmed that 400 °C is the optimum temperature for the largest furfural yield (~65 mol%), based on the doping of 1.1 mol% Pd on ZnSO₄ and a catalyst to allose mass ratio of four. Subsequently, by fixing the reaction temperature at 400 °C and the catalyst to allose mass ratio of four, we confirmed the largest furfural yield at the Pd doping amount of 1.1 mol% in Fig. S6b. The doping of less Pd, in particular around 0.4 mol% is insufficient for the removal of the formaldehyde side group, whereas the doping of more than 1.1 mol% Pd probably caused the secondary reactions of volatiles for over-cracking, and therefore, decrease in the furfural yield. Finally, by

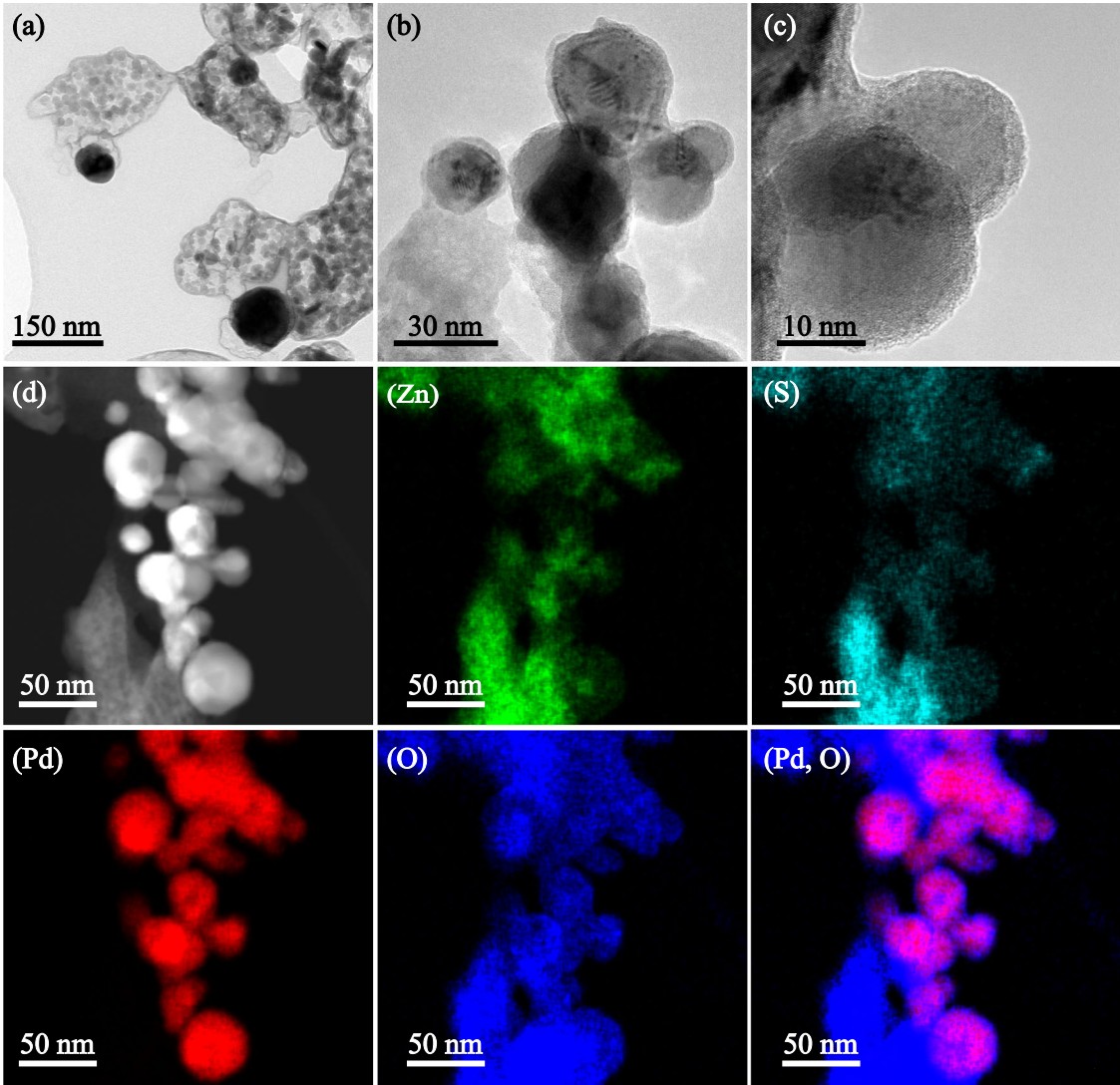

**Fig. 3 | TEM observation of 1.1 mol% Pd-doped ZnSO₄ catalyst.** Panel **a** for the low-magnification TEM picture. Panels **b**, **c** for the high-magnification TEM for Pd clusters and individual Pd particles. Panel **d** for the STEM observation of catalyst and its mapping of individual elements including Zn, S, Pd, O, and the superimposition of Pd and O (i.e., Pd-O).

varying the catalyst to mass ratio, the furfural yield was further improved to around 74 mol% at the catalyst to biomass mass ratio of eight, as demonstrated in Fig. S6c.

With the use of these optimized experimental conditions, we subsequently tested three monomers, C5 xylan, and C6 allose and glucose. To reiterate, the Pd-PdO/ZnSO₄ catalyst refers to a nominal 1.1 mol% Pd hereafter. The ZnSO₄ support only and pure PdO were also tested for comparison. The blank tests were further run as the baseline to demonstrate the effect of the three catalysts. Regarding the selectivity and yield of furfural in the blank cases (Fig. 4a), the furfural yield was ~40 mol% for C5 xylan and ~20 mol% for the two C6 sugar monomers. The selectivity of furfural is also relatively low for all the three blank cases. Nevertheless, the higher furfural yield from C5 sugar agrees with the commercial hydrolysis process using C5 sugar as the feedstock for furfural production. In addition, since the permanent gases were negligible from the pyrolysis of C5 xylan alone (Fig. 4a'), it is indicative that dehydration is the principal reaction for its conversion to furfural, as expected. In contrast, for both C6 sugar monomers, around 0.1–0.2 mol/mol·sugar was produced for CO and $CO_2$ each, indicative of a small extent of the decomposition of C6 sugars upon thermal shock.

The use of pure PdO catalyst in Fig. 4b led to a decrease in the yield of furfural, in particular in the case of C6 glucose. This should be related to its strong catalytic effect with the abundance of strong acid sites (Table 1), leading to the over-cracking of C-C bonds in the sugar rings for the production of large quantities of permanent gases including $H_2$, CO and $CO_2$ in Fig. 4b'. In contrast, for the ZnSO₄ support alone in Fig. 4c, the use of it for C5 xylan improved the furfural yield to ~75 mol%. This is consistent with the results for other sulfates for the pyrolysis or solvolysis of C5 sugars elsewhere[26–32]. The yields of permanent gases in Fig. 4c' are also negligible, supporting the strong dehydration effect of the ZnSO₄ support. For the C6 sugar monomers, here again, the use of ZnSO₄ alone is beneficial in improving the yield and selectivity of furfural. Indeed, we have screened nine different metal oxides and sulfates, and have confirmed that ZnSO₄ has the highest yield, as evidenced in Fig. S8. Nevertheless, the use of the Pd-PdO/ZnSO₄ is more pronounced, boosting the furfural yield to 74.0 mol% for allose (spectra in Fig. S9) and 77.3 mol% for glucose in Fig. 4d. Such yields are remarkably higher than the highest furfural yields reported for the solvolysis method[41], 24.5 mol% from glucose in the presence of $Fe_2(SO_4)_3$ as the catalyst, and 14.3 mol% from cellulose with $FeCl_3 \cdot 6H_2O$ as the

**Table 1 | Physiochemical properties of different catalysts**

| | Pd ratio, mol% | Acidity, by $NH_3$-TPD, mmol/g | | | | Acidity, by Pyridine-FTIR | Porosity analysis | | | |
|---|---|---|---|---|---|---|---|---|---|---|
| | | Weak (<200 °C) | Medium (200–600 °C) | Strong (600–800 °C) | Total | B/L[a] | $S_{total}$,[b] $(m^2/g)$ | $S_{micro}$,[c] $(m^2/g)$ | V,[d] $(cm^3/g)$ | Average pore size[e], nm |
| Pd-PdO/ ZnSO$_4$[f] | 1.1 | 8.6 | 5.2 | 0.3 | 14.1 | 0.25 | 4.49 | 0.61 | 0.0064 | 5.7 |
| PdO | 100 | 0 | 0.2 | 4.8 | 5.0 | 0.17 | 2.02 | 0.07 | 0.0028 | 5.5 |
| ZnSO$_4$ | 0 | 7.8 | 5.3 | 0.2 | 13.3 | 0.11 | 3.51 | 0.20 | 0.0049 | 5.6 |

[a]Brønsted acid/Lewis acid, semiquantified by area%; [b]BET surface area; [c]T-plot micropore area; [d]Single point adsorption total pore volume of pores less than 206.503 Å width at p/p° = 0.900000000; [e]4 V/A by BET; [f]1.1 mol% Pd-doped ZnSO$_4$.

catalyst. The solid residue yield is also the lowest with Pd-PdO/ZnSO$_4$ catalyst, at 9.3 wt%, compared to 13.9 wt% with the ZnSO$_4$ catalyst and 32.5 wt% for the blank case. Clearly, the combination of Pd and ZnSO$_4$ is capable of dehydrating as well as selectively cleaving the side group formaldehyde (HCHO) rather than over-cracking the entire C6 sugar molecule. Moreover, the fragmented side group mostly transformed into permanent gases via decomposition and steam reforming, as evidenced by the formation of around 0.8 mol (CO + CO$_2$) /mol-C6 sugar (see Fig. 4d') that matches well with the presence of one mole of formaldehyde per mole of C6 sugar molecule, and a 77.3 mol% yield of furfural in Fig. 4d. These results also support a satisfactory mass balance from the micro-reactors used here.

The superior activity of the Pd-PdO/ZnSO$_4$ catalyst was further confirmed by testing cellulose, both dry and wet in Pyro-probe, as shown in Figs. 5a–d with the original spectra presented in Figs. S10a–d. For dry cellulose without the addition of catalysts in Fig. 5a, its furfural yield reached ~15%, which is close to the results for its two monomers in Fig. 4a. The addition of Pd-PdO/ZnSO$_4$ improved the furfural yield to ~45%. Nevertheless, this yield is still lower than the furfural yields (60–80%) for its two monomers in Fig. 4c. Interestingly, upon testing the wet cellulose containing 10 wt% moisture, we noticed a remarkable increase in the yield of furfural for all cases. In particular, with the use of the Pd-PdO/ZnSO$_4$ catalyst, the furfural yield reached ~75%, which is clearly in line with the result for its glucose monomer. The TGA-MS results of dry and wet cellulose without catalyst in Fig. S10e show that pyrolysis of the wet cellulose alone induced the release of the gases, including H$_2$, CO, CO$_2$ and HCHO, at a relatively lower temperature than the case with dry cellulose as the feedstock. Particularly, the formation of the intermediate LGO was prohibited during wet cellulose pyrolysis. This is due to the initial hydrolysis of wet cellulose, cleaving the polymeric structure into monomers that are relatively easy to further decompose. In particular, the Brønsted acid within a catalyst is beneficial for the hydrolysis of cellulose[51]. Finally, regarding the permanent gases in Fig. 5a'–d', the use of the Pd-PdO/ZnSO$_4$ catalyst for wet cellulose produced the largest quantity of individual and total gases. This further supports a promoted cleavage of side groups and their reforming into syn-gas by this catalyst.

Moreover, the lab-scale fixed-bed reactor based on the use of ~6 g glucose validated the superiority of the Pd-PdO/ZnSO$_4$ catalyst and the importance of moisture within the feedstock, as shown in Fig. S11 and Table S1. Upon the use of wet glucose containing 10 wt% moisture, the furfural yield was improved from 4.2 mol% for the blank case with no catalyst to 62.4 mol% and 76.6 mol% for the mass ratio of catalyst to glucose at one and four, respectively. These results are comparable with the Pyro-probe results in Fig. 4d. Finally, the high activity of the catalyst was validated by the testing of two real biomass feedstocks, corncob and sugarcane bagasse, with the results summarized in Table 2 and Fig. S12. Under the optimum conditions based on pure allose, we achieved a furfural yield of 32.9 wt% on the mass basis of wet corncob, and 18.3 wt% for wet sugarcane bagasse. A higher furfural yield from corncob than that from sugarcane bagasse shall be

contributed to the larger content of cellulose and hemicellulose content in corncob[52]. Nevertheless, both yields from corncob and sugarcane bagasse are remarkably larger than the commercial furfural yield of 8–12 wt%[53]. Additionally, a good stability and durability were confirmed by the repeated testing of the Pd-PdO/ZnSO$_4$ with glucose as the feedstock in Fig. S13. Despite a slight decrease in the second cycle, the furfural yield and selectivity remained relatively stable afterwards. In addition, the S$_{2p}$ XPS analysis results in Figs. S14a and a' confirmed an identical peak position and width between the fresh and spent Pd-PdO/ZnSO$_4$ catalysts, proving the thermal stability of this sulfate-supported catalyst. In contrast, the ZnSO$_4$ support alone is rather unstable, displaying a broadened S$_{2p}$ XPS spectrum with the formation of new species shown in Figs. S14b and b'. Inferably, the unique structure in Figs. 2–3 is accountable.

## Reaction pathways of C6 sugars to furfural

For the pyrolysis of blank cellulose at 400 °C, the results in Fig. S10 confirmed the preferential formation of C6-bearing liquid products, including 5-HMF, LGO and a small amount of 1,4:3,6-dianhydro-alpha-d-glucopyranose (DGP, C$_6$H$_8$O$_4$). This broadly agrees with the general transformation route of C6 sugars in Fig. 1b. In addition, another C6 product, LGA, with a selectivity of 54.1% was confirmed from the pyrolysis of glucose alone in the fixed-bed reactor experiments carried out by us, as shown in Fig. S11a. Indeed, LGA has been frequently reported as one of the principal products from the pyrolysis of glucose or cellulose in fixed-bed reactors[18,54,55]. DFT modeling in a previous study also suggested that LGA is the most kinetically favored product from the dehydration of glucose[56].

Based on these observations, the three principal C6 intermediates, LGA, LGO and 5-HMF, were tested separately in Pyroprobe, with and without the ZnSO$_4$ support and Pd-PdO/ZnSO$_4$ catalyst at a catalyst to biomass mass ratio of eight. The summarized results and respective GC-MS spectra can be found in Figs. S15 and S16 of the SI. As has been confirmed, pyrolysis of the dry LGA alone confirmed its high stability at 400 °C, with an extremely low conversion of approximately 11% (see Fig. S15a) and the principal formation of LGO and 5-HMF, which are accompanied by furfural in a tiny amount. However, the use of ZnSO$_4$ support alone boosted the conversion of LGA to ~72%, along with the formation of furfural and the remarkable inhibition of the formation of 5-HMF. The use of the Pd-PdO/ZnSO$_4$ catalyst further enhanced the selectivity of furfural over LGO. In particular, with the addition of the Pd-PdO/ZnSO$_4$ catalyst, the overall conversion of wet LGA was almost complete, and the selectivity of furfural reached ~61%, relative to only approximately 20% for LGO, as shown in Fig. S15b with the spectra presented in Fig. S16a. Similar to that observed for the wet cellulose in Fig. 5a–d, initial hydrolysis of LGA should also occur on the Brønsted (H$^+$) or Lewis acid site of the catalyst, leading to the formation of glucose that is more readily catalyzed. Regarding the other two C6 products, LGO and 5-HMF, the results in Figs. S16b and c confirmed the lowest conversion to furfural, irrespective of the presence and type of catalyst. Indeed, a previous DFT study also reported the difficulty of

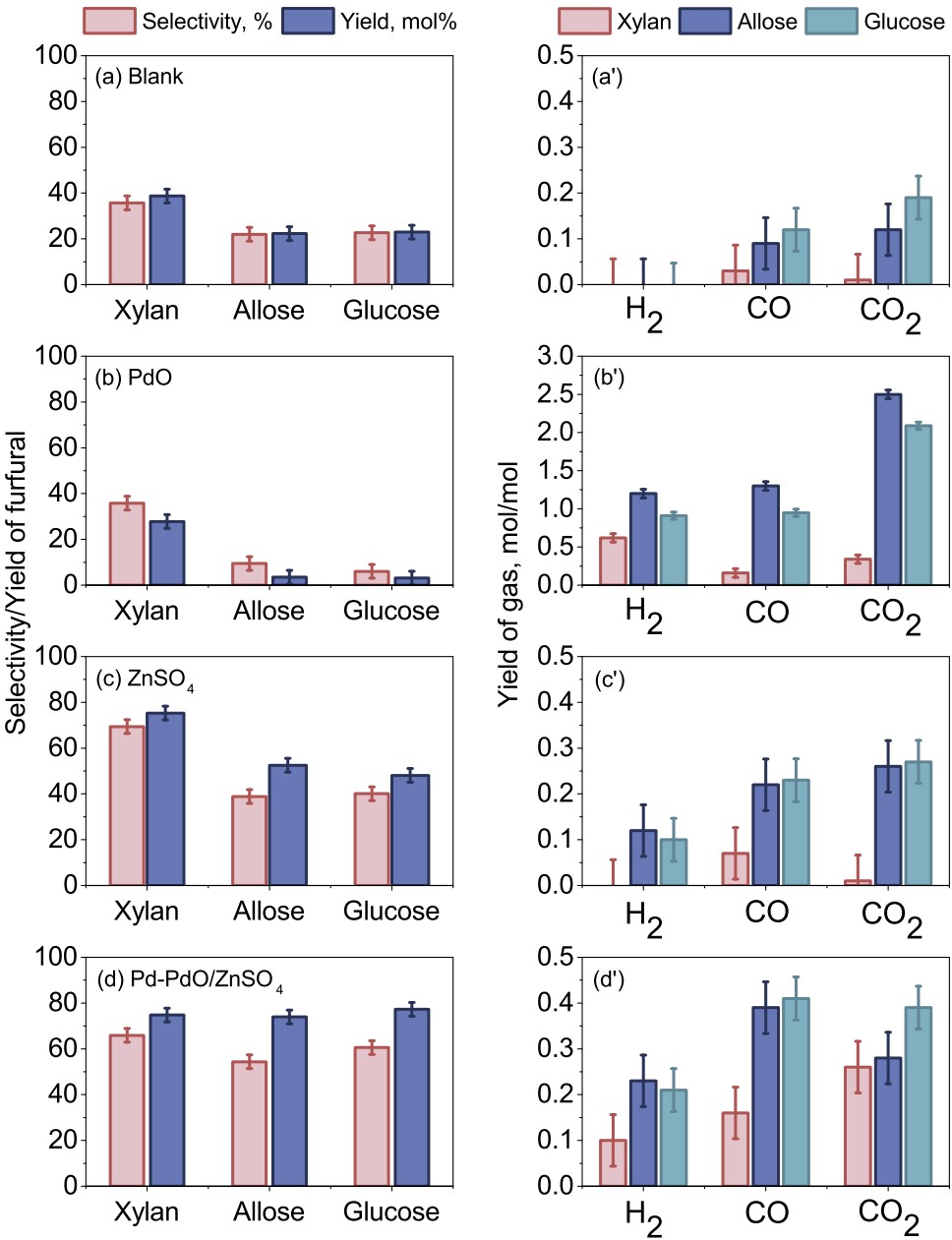

**Fig. 4 | Product distribution from the pyrolysis of C5 xylan, C6 allose and C6 glucose at 400 °C.** Panels **a** and **a'** for the selectivity and yield of furfural, and the yields of individual gases from the blank tests, respectively. Panels **b** and **b'** for the respective results from the PdO catalyst. Panels **c** and **c'** for the respective results from the ZnSO$_4$ catalyst. Panels **d** and **d'** for the results from the Pd-PdO/ZnSO$_4$. All the catalysts were tested at a mass ratio of eight to the feedstock, and the error bars were based on the standard deviation of the results from more than three repetitions.

breaking the hydroxymethyl group (-CH$_2$OH) from 5-HMF to form furfural[57]. Based on this evidence, it is concluded that the catalyst we developed greatly suppressed the formation of LGO and 5-HMF from C6 sugars, while LGA should be the principal precursor for the formation of furfural from C6 sugars.

Next, to confirm which carbon (C) was removed from the C6 sugar ring leading to the formation of furfural, $^{13}$C-labeled glucose at different positions and its unlabeled reference were tested by Py-GC-MS, with the results summarized in Figs. 6a and S17. For the pyrolysis of the unlabeled glucose in Fig. 6a, furfural (C$_5$H$_4$O$_2$, MW = 96) obtained has a mass/charge ratio (m/z) of 96 for its most intense peak with an arbitrary height of 100 in the mass spectrum (see Fig. S17). Upon testing of the $^{13}$C-123-labeled glucose, the m/z position for the most intense peak of furfural was increased to 99, indicating that the three carbons located at positions 1, 2, and 3 were retained in the furan ring of

furfural. The testing of $^{13}$C-1-labeled glucose also confirmed the appearance of m/z = 97 for the most intense peak. In contrast, when $^{13}$C-456-labeled glucose was tested, the most intense peak of furfural appeared at m/z = 98, suggesting that one of the $^{13}$C carbons in the $^{13}$C-456-labeled glucose was removed from the glucose ring. More specifically, it is from the $^{13}$C-6-labeled position as its most intense peak was reverted to 96, relative to the m/z position of 97 for the position of $^{13}$C-5. In other words, the carbon at the C-6 position was removed from the glucose ring, while those at the other five positions remained intact and transferred to the furfural molecule.

In addition, considering that formaldehyde (HCHO, MW = 30) is the primary gaseous product released during the conversion of glucose to furfural, its formation was then traced by testing $^{13}$C-1, $^{13}$C-5 and $^{13}$C-6-labeled glucose in TGA-MS, with the results shown in Figs. 6b-d and S18. The mass spectrum of unlabeled formaldehyde has three

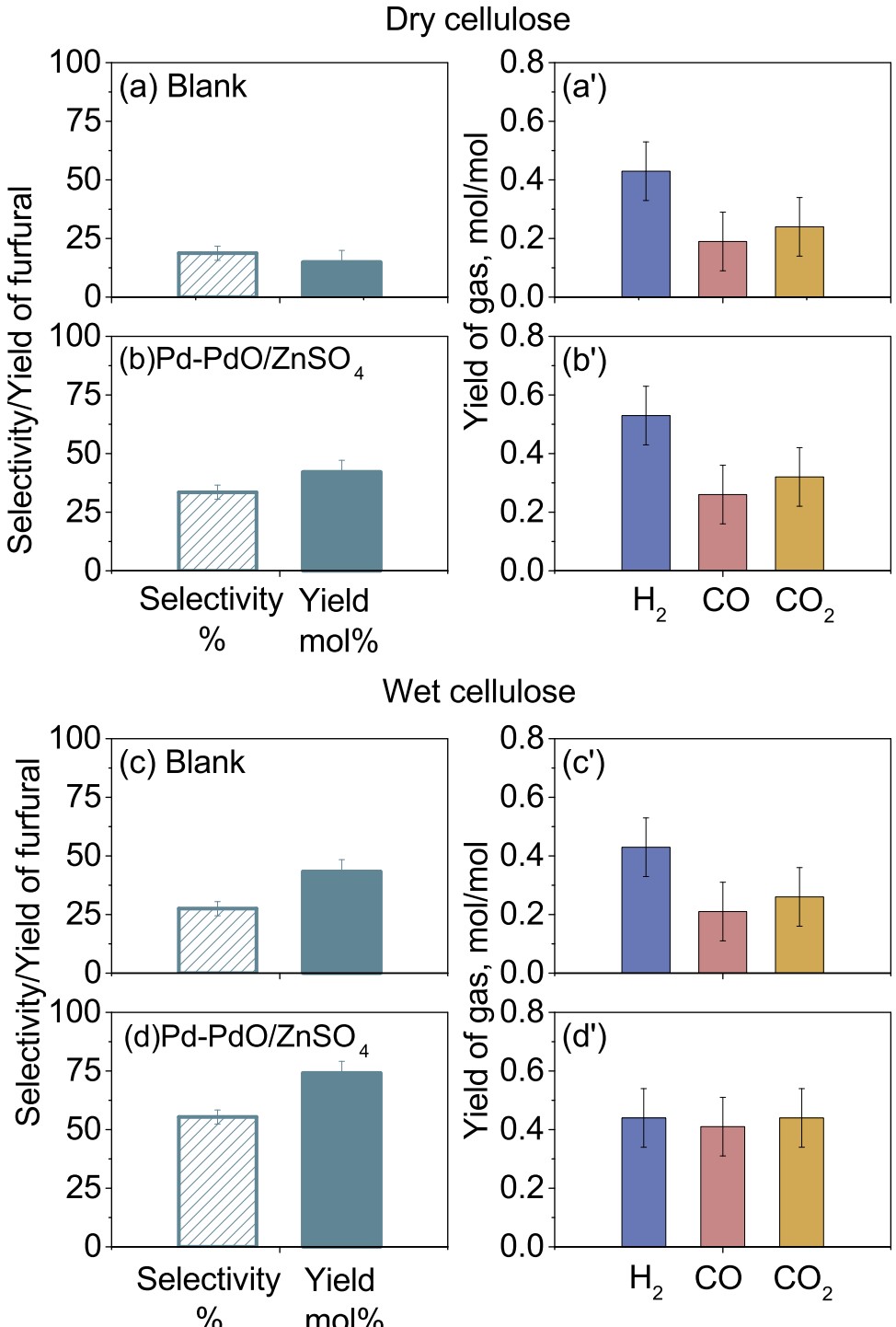

**Fig. 5 | Product distribution from the pyrolysis of dry and wet cellulose.** Panels **a** and **a'** are for the selectivity and yield of furfural, and the yields of individual gases from dry cellulose without catalysts, respectively; Panels **b** and **b'** are for the respective results from dry cellulose with the Pd-PdO/ZnSO$_4$ catalyst; Panels **c** and **c'** are from wet cellulose without catalysts; and Panels **d** and **d'** are for the wet cellulose with the Pd-PdO/ZnSO$_4$ catalyst. All the catalysts were tested at a mass ratio of eight to the feedstock, and the error bars were based on the standard deviation of the results from more than three repetitions.

major peaks at m/z = 28 (arbitrary height of 24), 29 (arbitrary height of 100) and 30 (arbitrary height of 58), as shown in Fig. S19. Here, the m/z of 30 for the unlabeled formaldehyde was traced, whereas the other two at 28 and 29 were not considered due to their overlap with CO (arbitrary height of 100 for m/z of 28 and 1 for m/z of 29) and even with CO$_2$, which also has peaks at m/z of 28 and 29. As demonstrated in Fig. 6b, $^{13}$C-6-labeled glucose is the only feedstock that promoted the peak intensity of m/z = 31 to a considerable extent, particularly in the presence of the Pd-PdO/ZnSO$_4$ catalyst. Furthermore, the time-resolved release profile in Figs. 6c, d confirmed the significant role of the catalyst in promoting the release of all the three m/z positions before 400 °C.

Based on all the experimental results above and previous studies[19,21,56–59], the overall reaction pathway for the conversion of C6 sugar to furfural is proposed in Fig. 7. Here, the initial hydrolysis of cellulose was excluded, as its occurrence on the Brønsted acid site is

**Table 2 | Product distribution from the pyrolysis of corncob and sugarcane bagasse with and without catalysts in Pyroprobe at 400 °C**

| | Corncob | | Sugarcane bagasse | |
|---|---|---|---|---|
| | $Y_{Furfural}$, %[a] | $S_{Furfural}$, %[b] | $Y_{Furfural}$, %[a] | $S_{Furfural}$, %[b] |
| Blank | 2.8 | 6.0 | 5.2 | 10.6 |
| $ZnSO_4$[c] | 8.9 | 15.9 | 13.0 | 27.2 |
| Pd-PdO/$ZnSO_4$, dry biomass[c] | 9.6 | 16.1 | 17.6 | 30.8 |
| Pd-PdO/$ZnSO_4$, wet biomass[c] | 32.9 | 58.8 | 18.3 | 32.5 |

[a]wt% based on dried biomass.
[b]area% in GC-MS.
[c]The catalyst to biomass mass ratio was fixed at eight.

well known. Steps 1–3 refer to the dehydration of glucose to LGA and its further dehydration to LGO or the direct dehydration of glucose to LGO upon thermal shock. However, in the presence of catalysts, these steps are significantly suppressed. Alternatively, steps 4–10 are enhanced remarkably, making furfural the principal product. In detail, step 4 refers to the hydrolysis of LGA, which is promoted by water/steam, as confirmed by the results of LGA as feedstock in Fig. S15b. Additionally, as the wet LGA has a furfural selectivity similar to that of dry glucose at ~61% (see Fig. 4d and Fig. S15b), it is concluded that step 4 should not be the rate control step for the overall conversion. Following the formation of glucose in step 4, a ring-opening reaction is required to create an acyclic form of glucose via step 5 for its conversion to C5 furfural, per the study on glucose elsewhere[60]. Next, the acyclic glucose should be isomerized via step 6, thereby enabling the fragmentation of formaldehyde from the C6 position. Consequently, the D-fructofuranose intermediate can be formed via step 7 by ring-closing at the C-2,5 positions. Afterwards, considering that the dehydration of glucose commences at 300 °C in Fig. 8a, whereas the appearance of formaldehyde only starts at 380 °C in Fig. 8c, it is inferable that the subsequent step 8 for Grob fragmentation likely follows the protonation of the hydroxyl group in D-fructofuranose, leading to the release of one free water molecule along with the free formaldehyde molecule. Afterwards, furfural is formed by the continued dehydration reaction of other hydroxyls in steps 9-10[19,21,56–59]. In addition, the free formaldehyde group formed in step 8 can further undergo decomposition and steam reforming reactions to turn into syn-gas via step 8', as evidenced by the results for three permanent gases, $H_2$, CO and $CO_2$ in Fig. 4 from Py-GC-TCD and in Figs. 8a–d from TGA-MS.

Finally, regarding the active sites on the developed Pd-PdO/$ZnSO_4$ catalyst, the catalytic effect of the $ZnSO_4$ support on the dehydration reactions throughout the entire reaction pathways in Fig. 7 is desired, as confirmed by the results for C5 sugars in Fig. 4a and by previous studies[32]. Nevertheless, it is still worth exploring the roles of metallic $Pd^0$, PdO and even $ZnSO_4$ in the critical Grob fragmentation from step 8 to step 10 for the formation of the final furfural. Therefore, DFT calculations were performed to construct the optimum geometry of the adsorbed D-fructofuranose, i.e., the key precursor formed prior to step 8 in Fig. 7, and the entire energy profile from the adsorption of D-fructofuranose to the final desorption of furfural product at −273.15 °C (Fig. S20) and 400 °C (Fig. 8i), which is the optimum reaction temperature in this work. Increasing the reaction temperature was found to be highly beneficial for the formation of furfural, as the overall reaction energy change (ΔE) decreases from 1.45 eV at −273.15 °C to −3.62 eV at 400 °C. Desorption of furfural is also much easier at 400 °C. The optimized surface models for $Pd^0$ (111), PdO (101) and $ZnSO_4$ (111) in Fig. S21 confirmed the presence of unsaturated Pd and Zn on the latter two surfaces. As a result, the adsorption of hydroxyl groups in

D-fructofuranose is stronger on the surfaces of $ZnSO_4$ and PdO than on metallic $Pd^0$, with surface adsorption energies of −0.22 eV for Pd, −0.33 eV for PdO, and −0.43 eV for $ZnSO_4$ at 400 °C (Fig. 8f–h). For the subsequent Grob fragmentation in step 8, the PdO surface should be the most active site, as it presents the lowest energy change of −1.29 eV, relative to −0.57 eV for metallic $Pd^0$ and −0.26 eV for $ZnSO_4$, as shown in Fig. 8i. In contrast, in the dehydration step 9, $ZnSO_4$ shows the lowest energy change of −1.05 eV relative to −0.78 eV for Pd and −0.70 eV for PdO. This is also desired and further confirms the important role of $ZnSO_4$ in dehydration. Regarding step 10 for the last dehydration reaction to form furfural, metallic $Pd^0$ is apparently the most active, with the lowest energy change of −2.34 eV. Finally, the desorption energy change for the formation of free furfural is only −0.45 eV on the PdO (101) surface relative to 0.32 eV on the Pd (111) surface and 0.52 eV on the $ZnSO_4$ (111) surface. This indicates a preferential and easy release of furfural from the PdO surface. These calculations support the presence of multiple active sites on our developed catalyst and its bifunctional properties. In addition, a strong synergy should exist between Pd and $ZnSO_4$ at their interface, thereby leading to the experimentally observed $Pd^0$-PdO core-shell structure, its moderate acidity and strong stability that are all accountable for the superior activity of the catalyst.

In summary, a novel, bifunctional and reusable Pd-PdO/$ZnSO_4$ catalyst with an optimum Pd loading of 1.1 mol% was designed to produce furfural from both C5 sugars and C6 sugars by flash pyrolysis at 400 °C. It is highly active in converting C5 and all kinds of C6 sugars from monomers to real sugarcane bagasse and corncob. More specifically, the use of wet feedstock is even beneficial in promoting the hydrolysis of cellulose and the LGA intermediate to glucose on the catalyst surface, thereby alleviating the necessity of predrying for biomass. Regarding the subsequent transformation of glucose, both its dehydration and Grob fragmentation of the side group formaldehyde for the selective formation of furfural were catalyzed selectively, whereas the formation of C6 by-products, including LGO and 5-HMF, was suppressed remarkably. Moreover, Grob fragmentation occurred at the C-6 position of glucose and mostly on the PdO (101) active site. The cleaved free formaldehyde molecules also underwent secondary reactions such as in-situ steam reforming to turn into green syn-gas. Regarding the $ZnSO_4$ support, it functioned as the active site for all the dehydration reactions as well as probably isomerization reactions. Additionally, metallic $Pd^0$ could be active in promoting the last dehydration step. A strong synergistic effect between the Pd and $ZnSO_4$ support balances the acidity and stabilizes the $SO_4^{2-}$ anion on the support too.

## Methods

### Feedstock

Raw materials, including cellulose, D-allose ($C_6H_{12}O_6$), D-glucose ($C_6H_{12}O_6$), xylan, 5-HMF ($C_6H_6O_3$), LGA ($C_6H_{10}O_5$) and LGO ($C_6H_6O_3$), were reagent grade with a purity >99% and were purchased from Merck Co Ltd. The chemicals for catalyst synthesis, including Pd($NO_3$)$_2$·$2H_2O$ (~40% Pd basis), $ZnSO_4$·$H_2O$ (≥99.9% trace metals basis) and $ZnSO_4$·$7H_2O$ (ACS reagent, 99%), were also purchased from Merck Co Ltd. The two real biomass feedstocks, corncob and sugarcane bagasse with particle sizes of 150–250 μm, were sourced from Oz Gun Mart in New South Wales, Australia and Sugarcane Juice Bar in Melbourne, Australia, respectively. They were dried in an oven at 105 °C for 15 h prior to the pyrolysis experiments.

### Catalyst synthesis

The anhydrous $ZnSO_4$ was obtained by the annealing of $ZnSO_4$·$H_2O$ in argon (Ar) at 550 °C for 2 h, with a heating rate of 10 °C/min. To prepare the Pd-laden $ZnSO_4$ catalysts, the classic wet impregnation method was employed. That is, a certain amount of Pd($NO_3$)$_2$·$2H_2O$ was mixed with 30 ml ethanol in a beaker and then combined with

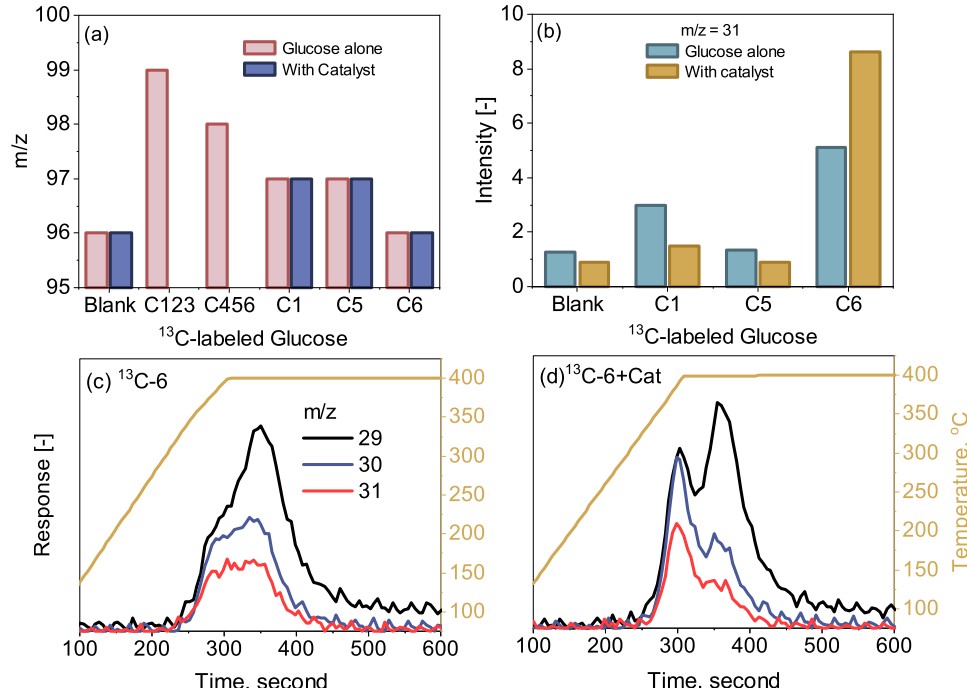

**Fig. 6 | MS detection of furfural and formaldehyde from the pyrolysis of unlabeled and ¹³C-labeled glucose samples.** Panel **a** for the m/z of furfural in Py-GC-MS. Panel **b** for the signal intensity of m/z = 31 from TGA-MS. Panel **c** for the TGA-MS signals at m/z of 29, 30 and 31 from the blank ¹³C-labeled glucose at the C-6 position. Panel **d** for the signals at the m/z of 29, 30 and 31 from the ¹³C-labeled glucose at C-6 blended with the Pd-PdO/ZnSO₄ catalyst. The catalyst was tested at a mass ratio of eight to the feedstock.

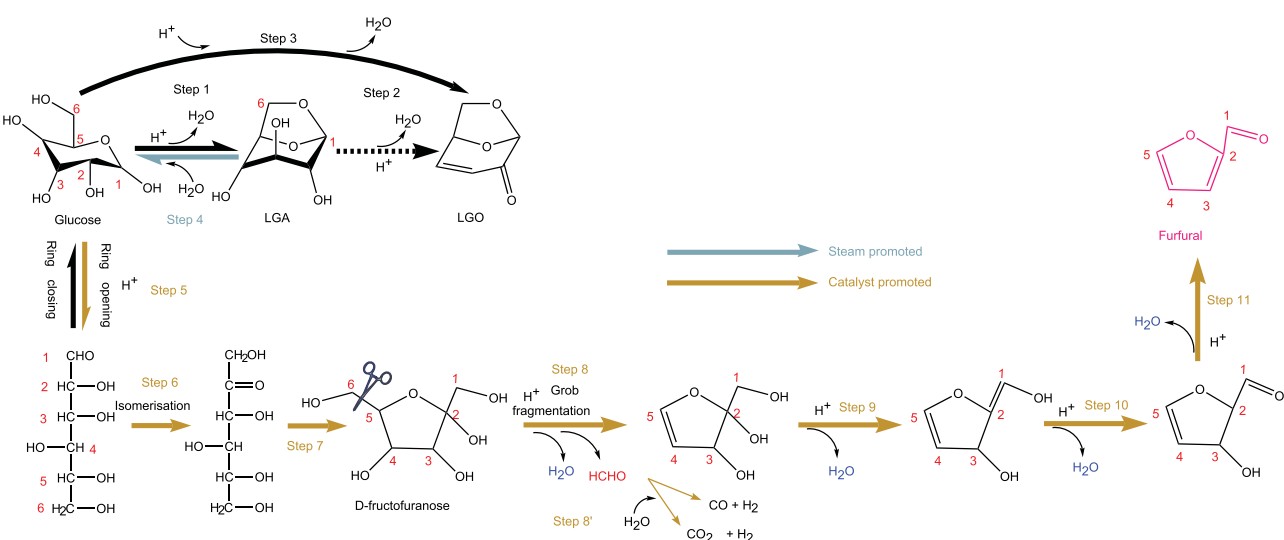

**Fig. 7 | Proposed reaction pathways for C6 sugar by catalytic pyrolysis of glucose upon the presence of Pd-PdO/ZnSO₄ and moisture/steam.** Steps 1–3 refer to dehydration of glucose to LGA and/or LGO upon thermal shock. Step 4 for the hydrolysis of LGA upon the promotion of added moisture from wet glucose. Step 5 for the creation of acyclic glucose via a ring-opening reaction. Step 6 for the isomerization of acyclic glucose. Step 7 for the formation of D-fructofuranose via the ring-closing reactions at C-2,5 positions. Step 8 for the Grob fragmentation reaction to release the formaldehyde molecule. Step 8′ for the in-situ steam reforming of formaldehyde molecule, and Steps 9-10 for the continued dehydration reactions to form the final product furfural.

ZnSO₄·7H₂O. The mixture was subsequently sonicated three times for 5 min each and stirred continuously with a glass rod to achieve thorough mixing. Afterwards, the solution was vacuum dried at 80 °C for 15 h, followed by annealing at 550 °C for 2 h in Ar. For catalysts with a nominal Pd content of 1.1 mol%, the resulting catalysts were labeled Pd-PdO/ZnSO₄. The composition of each catalyst was quantified by inductively coupled plasma-optical emission spectroscopy (ICP-OES, Perkin Elmer Optima 7000DV). Similarly, a pure PdO catalyst was

synthesized by the annealing of Pd(NO₃)₂·2H₂O under the same conditions as Pd-PdO/ZnSO₄.

## Pyrolysis experiments
The catalysts were tested in a Pyro-probe microreactor coupled with GC-TCD/FID/MS detectors. The system houses a fixed-bed quartz tube reactor of 2 mm in diameter and 30 mm in length, which is the same as that reported in our previous study[61]. In brief, for each run, a biomass

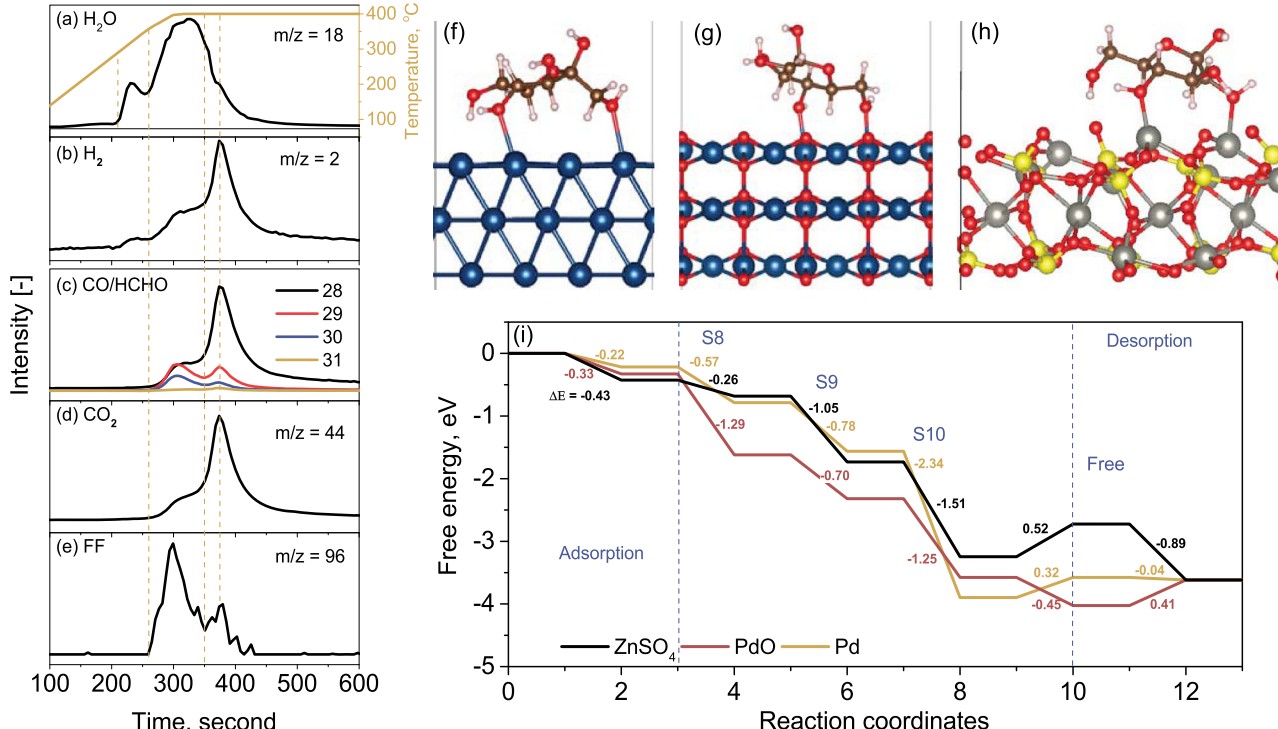

**Fig. 8 | TGA-MS profiles of the principal products from and DFT calculation results for the pyrolysis of glucose.** Panels **a**–**e** for the TGA-MS spectra of the products obtained from pyrolysis of glucose, including $H_2O$ (**a**), $H_2$ (**b**), CO and HCHO (**c**), $CO_2$ (**d**) and furfural (FF) (**e**). Panels **f**–**h** for the optimized geometries for the adsorption of D-fructofuranose on Pd (111), PdO (101), and $ZnSO_4$ (111), respectively. Panel **i** for the free energy changes for the formation of furfural on the surfaces of Pd, PdO, and $ZnSO_4$ along the reaction coordinate at 400 °C. S8-S10 in panel (**i**) refer to the reaction steps No 8–10 specified in Fig. 7.

of approximately 0.35 (±0.02) mg was placed in the first stage of the quartz tube, the volatiles produced from which were carried by the purge gas (helium) into the second stage where the catalyst was located, and the biomass and catalyst were secured by the quartz wool at each end. The tests were run at 300–500 °C at a heating rate of ~140 °C/s[49] with a flow rate of 26 mL/min and a duration of 25 s at each set temperature to ensure complete conversion of biomass. In some experiments, 0.25–1.40 μL water was added to the reactor in an amount equivalent to the content of chemical water in $ZnSO_4 \cdot 7H_2O$, unless otherwise stated. The effluents discharged were monitored online by a TCD detector (Agilent, 7890 B) coupled with a Molseive column and a Hayesep Q column for noncondensable gases, including $H_2$, $CH_4$, CO, $CO_2$, $C_2H_4$, $C_2H_6$ and $H_2O$, and an FID detector (Agilent, 7890 B) coupled with a capillary column and an MS detector (Agilent 5977B) for liquid products. The sample loop size of the gas product was 10 mL. The peak assignments in FID or MS patterns are summarized in Table S2. At least two repeat experiments were carried out for all the conditions studied here.

The solid yield $Y_{mass}(solid)$ was determined by weighing the feedstock and the solid residue after reaction using a microscale balance with an accuracy of 1 μg, per Eq. (1). The yield of each gas, $CO_2$, CO and $H_2$, was calculated by its calibration curve, and thus, the total gas yield, $Y_{mass}(gas)$, can be obtained by Eq. (2) based on the calibrated molar yield of each gas. The calibration details can be found in Section 5 of the SI. Due to the complex composition of bio-oil, it is nearly infeasible to calibrate all the compounds. Therefore, the liquid yield $Y_{mass}(liquid)$ was determined by the mass difference between feedstock and gas and solid, as shown in Eq. (3). The calibration of pure furfural confirmed that the error of the liquid yield is approximately 3%. Details of the mass balance in the Pyro-probe reactor can be found in Section 3 of the SI.

The selectivity of furfural, $S(furfural)$, and of LGO, $S(LGO)$, were determined by their area percentages in the GC-MS spectra Eqs. (4, 5),

which is linear to their mass concentrations[36]. For the target product furfural, a detailed calibration was performed to determine its mass concentration, $S_{mass}(furfural)$, based on its area selectivity, $S(furfural)$. The calibration method is presented in Section 5 of the SI, with the main results summarized in Table S3 and Figs. S22, S23. Afterwards, the furfural yield, $Y_{mass}(furfural)$, can be calculated based on the total liquid yield and furfural concentration, as shown in Eq. (6).

$$Y_{mass}(solid) = \frac{m_{solid}}{m_{feedstock}} \times 100\% \quad (1)$$

$$Y_{mass}(gas) = \left( \sum Y_{mol}(x) \times M(x) \right) / M_{feedstock} \quad (2)$$

$$Y_{mass}(liquid) = 100 - Y(gas) - Y(solid) \quad (3)$$

where the symbol $Y$ refers to the mass or molar yield, m is the mass, M is the molecular weight, and x stands for $CO_2$, CO, or $H_2$.

$$S(furfural) = A(furfural)/A_{total} \quad (4)$$

$$S(LGO) = A(LGO)/A_{total} \quad (5)$$

$$Y_{mass}(furfural) = Y_{mass}(liquid) \times S_{mass}(furfural) \quad (6)$$

where the symbol $S$ represents the selectivity of furfural or LGO, and $A$ represents the peak area in the GC spectra. $S_{mass}(furfural)$ was determined by the calibration curve in Fig. S23 of the SI.

To validate the performance of the catalysts on a larger scale, experiments were also conducted in a lab-scale horizontal furnace with a two-stage fixed-bed quartz tube reactor (50 mm ID and 600 mm length) using 200 mL/min $N_2$ as the carrier gas.

Approximately 6 g of biomass was loaded into a crucible sample holder with and without the addition of a catalyst, and then the sample holder was inserted into the preheated furnace at 400 °C. For the testing of wet biomass, 10 wt% water on the mass basis of glucose was mixed with glucose. The liquid bio-oil was collected by an impinger train, as shown in Fig. S25, and then analyzed by a gas chromatography-mass spectrometer (GC-MS, HP6890). For quantification, the yield of solid, $Y_{mass}(solid)$, was determined by weighing the feedstock and the solid residue, per Eq. (7). The yield of liquid was determined by the mass difference of the impinger train before and after the reaction, as shown in Eq. (8). The permanent gases, CO, $CO_2$ and $H_2$, were analyzed by a gas analyzer (Sensotec Rapidox 5100). The total gas yield can be determined by Eq. (9). Details for the pyrolysis experiment in the fixed-bed reactor can be found in Sections 4 and 5 of the SI.

$$Y_{mass}(solid) = \frac{m_{solid}}{m_{feedstock}} \times 100\% \tag{7}$$

$$Y_{mass}(liquid) = \frac{m_{collector+liquid} - m_{empty\,collector}}{m_{feedstock}} \times 100\% \tag{8}$$

$$Y_{mass}(gas) = (m(CO_2) + m(CO) + m(H_2))/m_{feedstock} \tag{9}$$

### TGA-MS analysis

To trace the variation in the mass to charge ratios (m/z) for the gases produced from the [13]C-labeled and unlabeled glucose, thermogravimetric analysis (TGA) was performed in an SDT 650 Simultaneous Thermal Analyzer coupled with a Pfeiffer Vacuum mass spectrometer (MS). For each run, approximately 10 mg of sample was heated from 50 °C to 400 °C in Ar at a heating rate of 80 °C/min and held for 5 mins at 400 °C. Volatiles from the TGA system were purged into the MS by Ar (100 mL/min) through a quartz capillary heated at 200 °C. The MS system was operated under a vacuum of $3.6 \times 10^{-6}$ hPa, and the characteristic fragment ion intensities of the volatiles were detected according to their respective m/z. The experiments were performed more than twice to ensure a high repeatability.

### Catalyst characterization

The acidity of the catalysts was measured by $NH_3$-temperature programmed desorption ($NH_3$-TPD) in a Micromeritics AutoChem II 2920. The $NH_3$ signal was recorded online by a TCD detector. The total acid site was semi-quantified by the peak area of $NH_3$ desorption. Infrared spectra (Pyridine-FTIR) of the catalysts were recorded with a Thermo (Nicolet 380) spectrometer in the wavelength range of 400–4000 cm$^{-1}$. More details can be found in Section 5 of the SI.

The X-ray absorption near-edge structure (XANES) analyses of Pd (K-edge) were undertaken on the X-ray absorption spectroscopy (XAS) beamline at the Australian Synchrotron. The data collection and interpretation processes were identical to our previous work[61]. In brief, the catalyst or standards were carefully mixed with cellulose to dilute the palladium content down to 100–200 ppm, after which approximately 50 mg of the mixture was pressed into a pellet for analysis. The Pd K-edge spectra of the catalyst, as well as the Pd foil and PdO standards, were recorded in fluorescence mode at 24,150–25,100 eV. The software Athena was used to extracting the information for the valence state of Pd. In addition, linear combination fitting (LCF) was performed on the Pd K-edge spectrum using pure Pd foil and PdO standards.

X-ray photoelectron spectroscopy (XPS) analyses of catalysts were conducted in a Thermo-Scientific Nexsa Surface Analysis Platform with a micro-focused, monochromated Al Kα source. All the spectra were calibrated by the carbon ($C_{1s}$) peak at 284.8 eV. The data were processed in Avantage software[62]. The reference binding energies of Pd, zinc (Zn) and sulfur (S) were defined based on the XPS database[63].

X-ray diffraction (XRD) analyses of the catalysts were conducted on a Rigaku MiniFlex 600 diffractometer operated at 40 kV and 15 mA with a Cu radiation source at 1.5418740 Å. High-temperature XRD (HT-XRD) analyses were performed in the Power Diffraction (PD) Beamline at the Australian Synchrotron. The PD beamline is located on a bending magnet source and operated at an energy of approximately 16 keV with an X-ray wavelength of 0.7752545 Å and a beam size of 1 mm (vertical) × 2 mm (horizontal). Peak identification was achieved by the search-match function in JADE and Match software. Information on the sample preparation and data collection procedure can be found in Section 5 of the SI.

For the morphologies and crystallization information related to Pd-bearing species in catalysts, transmission electron microscopy - selected area electron diffraction (TEM-SAED) and high-resolution TEM (HRTEM) were conducted on an FEI Tecnai G2 T20 TWIN TEM with an accelerating voltage of 200 kV. Scanning transmission electron microscopy (STEM) was conducted on an FEI Tecnai G2 F20 S-TWIN STEM. Details for the sample preparation procedure can be found in Section 5 of the SI.

### Computational methods

Bulk models of Pd, PdO and $ZnSO_4$ were retrieved from the Materials Project database[64]. The (111) facet of Pd, (101) facet of PdO, and (111) facet of $ZnSO_4$ were selected based on the experimental observation of these facets by TEM (Fig. 2f, g) and the fact that these facets have the highest intensity peaks in the XRD patterns (Figs. S1, S2). It was also broadly reported in previous studies that Pd (111)[65,66] and PdO (101)[67,68] were used for furfural hydrogenation reactions. To avoid interactions from the adjacent images, the lateral sizes of these slab supercells were constructed to be larger than 10 Å, and their lattice constants along the $z$-direction were fixed at 30 Å. Accordingly, the slab supercells of Pd, PdO, and $ZnSO_4$ contain 48, 96, and 91 atoms, respectively. All computations were performed using spin-polarized density functional theory (DFT) calculations via the Vienna ab initio simulation package (VASP)[69,70]. The generalized gradient approximation in the form of Perdew-Burke-Ernzerhof was used to describe the exchange-correlation interaction for electrons[71]. Due to the large sizes of these slab models, the Brillouin zone sampling k-mesh was set as $1 \times 1 \times 1$, and the energy cutoff was set as 450 eV. The energy and force convergence criteria were set as $1 \times 10^{-4}$ eV and 0.02 eV/Å, respectively. The van der Waals interaction as parameterized by Grimme (DFT-D3) was included during all the computations[72,73]. Dipole correction along the $z$-direction was included for slab calculations. The free energy change for each reaction step at a finite temperature $T$ is calculated as $\triangle G = \sum G^T_{products} - \sum G^T_{reactants}$. The free energy of each species at temperature $T$ ($G^T$) is expressed as $G^T = E + U_{0 \rightarrow T} + E_{ZPE} - TS$, where $E$ is the total energy, $U_{0 \rightarrow T}$ is the inner energy changed from 0 K to $T$, $E_{ZPE}$ is the zero-point energy, and $S$ is the entropy. The entropy of each reaction intermediate was calculated by fixing the surface slab. The VASPKIT code was used for postprocessing of the VASP calculated data[74].

## Data availability

The main data generated in this study are provided in the Supplementary Information. The source data used in this study are available in the Figshare database (https://figshare.com) under the accession code of https://doi.org/10.6084/m9.figshare.22186537.

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

## Acknowledgements

This work was funded by the Australian Research Council (ARC) Linkage Project (180100128) received by L.Z. The HTXRD and XAS analyses were undertaken on the Powder Diffraction beamline (10BM1) and X-ray Absorption Spectroscopy beamline under the Australian Synchrotron beamtime awards of 17119 (L.Z.) and 18547 (Q.Z.), respectively. The first author would like to acknowledge the China Scholarship Council (CSC) for her Ph.D. living allowance support. Dr. Tim Williams at the Monash Centre of Electron Microscopy (MCEM), Monash University, is acknowledged for the TEM analysis. Monash X-ray Platform (MXP) is also acknowledged for the XPS analysis. Dr. Jianghao Zhang in the Research Centre for Eco-Environmental Sciences, Chinese Academy of Sciences, was acknowledged for conducting the pyridine-FTIR analyses.

## Author contributions

Q.Z. conceived the research ideas, designed and carried out the experiments, and wrote the manuscript. J.G. performed the DFT modelling and wrote the modelling section of the manuscript. J.W. carried out some of the Pyro-probe experiments and TGA-MS analyses. A.D. helped with the XRD, TGA-MS, and NH₃-TPD characterizations. S.Y. helped with the Pyro-GC-MS analyses. L.Z. conceived the research ideas, supervised and led the project, and revised the manuscript. All authors discussed the results and contributed towards data interpretation and commented on the manuscript.

## Competing interests

The authors declare no competing interests.
