## [Peer Review File · Nature Communications]

High Production of Furfural by Flash Pyrolysis of C6 Sugars and Lignocellulose by Pd-PdO/ZnSO₄ CatalystREVIEWER COMMENTS

Reviewer #1 (Remarks to the Author):

The manuscript NCOMMS-22-43531-T by Zhang and coworkers reported a Pd-PdO/ZnSO₄ catalyst can selectively enable the formation of furfural from C₆ sugars (such as allose, glucose, and cellulose) via flash pyrolysis. This transformation has been studied previously with a series of catalysts, and current catalyst showed a better result than a previous report by using zeolites (ref. 23). However, I could not recommend this manuscript to Nature Communication currently based on following concerns.

1. Several Pd-PdO/ZnSO₄ catalysts with different Pd loading were prepared and employed in this study. However, the choice and dose of the catalysts are chaotic. For example: 1) In figure 4, the catalyst to feedstock mass ratios for C₅ xylan and C₆ glucose were eight, and it was changed as four for C₆ allose. In figure 5 and Table S2, the catalyst/biomass ratios for cellulose and LGA were not presented. 2) Which catalyst was used for pyrolysis of allose, glucose, LGA and cellulose? It seems that the authors chose Pd-PdO/ZnSO₄ (Pd 1.1%), but with no any reasons.
2. For corncob and sugarcane bagasse: why Pd-PdO/ZnSO₄-0.4 mol% show better performance than Pd-PdO/ZnSO₄-1.1 mol%? (Table 2). As comparison, Pd-PdO/ZnSO₄-1.1 mol% is better than Pd-PdO/ZnSO₄-0.4 mol% for allose pyrolysis (Fig S6).
3. The proposed reaction pathways suggested that glucose should be the intermediate for the transformation of LGA to FF. Why FF yields from LGA are higher than that from glucose?
4. In 4.2 Catalyst synthesis, what is the propose of the annealing of ZnSO₄·H₂O to anhydrous ZnSO₄?
5. The catalyst was recycled by a tweeter, how about the recovery rate? The characterizations of recycled catalyst were not described.

Reviewer #2 (Remarks to the Author):

This is an impressive work of Zhang and co-workers on the production of Furfural by means of Flash Pyrolysis starting from C₆ Sugars promoted by the Pd-PdO/ZnSO₄ Catalyst.

The research area is of high relevance, being timely and a hot topic at the moment.

In general, the paper is very solid: experiments are well conducted, a full physico-chemical characterization of the catalyst was carried out and conclusions are consistent with experimental data. In my opinion, paper can be accepted almost as such even for a "big journal" such as Nature Communication.

Some minor point:

- Maybe the title is misleading. Maybe it is better to change into: "High Production of Furfural by Flash Pyrolysis of C₆ Sugars and Lignocellulosic Biomasses promoted by Pd-PdO/ZnSO₄ Catalyst"
- some english polishing to correct typos and mistakes is needed.

Otherwise, a very nice paper.

Reviewer #3 (Remarks to the Author):

The manuscript discusses the use of Pd supported on ZnSO₄ for converting carbohydrate pyrolysis vapors to furfural. Furfural is an important product molecule and the apparent ability of the Pd/ZnSO₄ to convert C₆ carbohydrate-derived molecules to furfural is interesting.

There are, however, several significant concerns with the manuscript:

1. The quantification of furfural production is based on just considering its content in the measured product stream. There are no overall mass balances, so even though the portion of the product stream that is furfural increases with the use of the Pd/SO₄ catalyst, it is not clear if the overall product stream is reduced, i.e., what is the portion of the reactant that ends up in the GC measured product stream for each scenario (blank, ZnSO₄, Pd/ZnSO₄). Clearly, the yield of furfural as determined from just the product stream is significantly different than the yield of furfural based on the amount of reactant converted. This is a significant shortcoming of the manuscript.
2. The definition of yield and selectivity is not at all clear in the main manuscript and requires digging into the supplementary information to find. As these definitions are seminal to understanding the reaction results, they must be clearly articulated in the main manuscript.
3. Almost all of the reaction results are given for 1.1 mol% Pd/ZnSO₄ (Table 2 being the only exception). Additionally, extensive characterization is only performed on the 1.1 mol% material. Therefore, why is the introduction claiming doping levels from 0.4-1.1 mol%? Additionally, the 0.4 and 1.1 mol% materials gave similar results in Table 2. Seems that the key point is Pd loading on ZnSO₄ leads to more furfural in the measured product and no statement is able to be made on what is a preferred Pd loading.
4. Lines 245-248 discuss the possible role of water in breaking the glycosidic bonds for the cellulose pyrolysis step. Why is the Zn⁺² result being discussed? In the pyroprobe experimental setup the Pd/ZnSO₄ catalyst is only in contact with the pyrolysis vapors.

Reviewer #1 (Remarks to the Author):

The manuscript NCOMMS-22-43531-T by Zhang and coworkers reported a Pd-PdO/ZnSO₄ catalyst can selectively enable the formation of furfural from C6 sugars (such as allose, glucose, and cellulose) via flash pyrolysis. This transformation has been studied previously with a series of catalysts, and current catalyst showed a better result than a previous report by using zeolites (ref. 23). However, I could not recommend this manuscript to Nature Communication currently based on following concerns.

Answer: Thank you very much for your comments. We highly appreciate your invaluable suggestions provided below. Per the comments of yours and from other reviewers, we have revised the paper significantly by conducting extra experiments to consolidate the conclusion, as well as grouping the results (in particular those in the supporting information, **SI**) to enable a fast track for the readers. All the revisions are highlighted in **blue** in the main manuscript and **SI**. The individual replies are further provided point by point below. We sincerely hope these accommodated revisions can address your concerns.

Comment 1: Several Pd-PdO/ZnSO₄ catalysts with different Pd loading were prepared and employed in this study. However, the choice and dose of the catalysts are chaotic. For example: **1-1)** In **Figure 4**, the catalyst to feedstock mass ratios for C5 xylan and C6 glucose were eight, and it was changed as four for C6 allose. **1-2)** In **Figure 5** and **Table S2**, the catalyst/biomass ratios for cellulose and LGA were not presented. **1-3)** Which catalyst was used for pyrolysis of allose, glucose, LGA and cellulose? It seems that the authors chose Pd-PdO/ZnSO₄ (Pd 1.1%), but with no any reasons.

Response 1: We sincerely apologize for providing massive yet messy (and sometimes unclear) information in the original manuscript. The primary rationale for having different (optimum) catalyst to feedstock mass ratios is based on the fact that we conducted the experiments for a large variety of feedstocks and a larger number of influential parameters including the catalyst to feedstock mass ratio. By the parametric testing of different feedstocks, we did notice the variation of the optimum catalyst to feedstock mass ratio. However, as the discussion on it would rather deviate the focus of the paper, we have decided to consolidate all the results for a fixed and identical mass ratio of eight, thereby remarkably simplifying the story while improving its clarity as well. Regarding the different optimum mass ratio for different feedstocks, we would like to make a separate paper for a detailed discussion in the near future.

Actions:

Overall revisions for the chaotic choice and dose of catalysts: To improve the clarity of the paper, we have not only fixed the catalyst to feedstock mass ratio at 8, but also made the following major revisions. As most of the comments are related to these changes, they are summarized below for a fast track.

- a) Consolidation of the paper for a simple (optimum) catalyst to feedstock mass ratio of eight. As such, extra experiments for feedstock at the fixed mass ratio of eight were run and added into the

paper.

- b) Streamlined the results into separate sections to ensure a good alignment of the results with the discussion in each section, and an easy match between the main manuscript and the SI. The revisions were mainly made for Sections 2.2 and 2.3, and in the SI, we have further categorised the results into different sections for the reviewers and readers to fast track.
- c) Addition of a separate paragraph and discussion in Section 2.2 for the parametric testing results, including different temperatures, different Pd loading amounts and different catalyst to biomass mass ratios. As such, the subsequent discussion on different feedstock can be easily followed up.
- d) Provision of detailed overall mass balance analysis for both Pyro-probe micro-reactor and Fixed-bed reactor in **SI**.
- e) Revision of the Experimental Section 4.3 to provide detailed calculation methods for product yield and selectivity.

Action per 1-1: Fig. 4 – Here again, we have rerun some experiments for the catalyst to feedstock (i.e. allose) mass ratio of eight to consolidate the results in this figure on **Page 15**. Additionally, we have revised the respective descriptions on **Lines 210-212, Page 13** (...*Finally, by varying the catalyst to mass ratio, the furfural yield was further improved to around 74 mol% at the catalyst to biomass mass ratio of eight, as demonstrated in Fig. S6(c)*); **Lines 214-218, Page 13**: (*With the use of these optimised experimental conditions, we subsequently tested three monomers, C5 xylan, and C6 allose and glucose, and compared the best catalyst, Pd-PdO/ZnSO₄ with 1.1 mol% Pd, with the ZnSO₄ support only and pure PdO at the catalyst to feedstock mass ratio of eight. The blank tests were also run as the baseline to demonstrate the effect of the three catalysts.*); and **Lines 254-255, Page 16: Fig. 1** (*Product distribution from the pyrolysis of C5 xylan, C6 allose and C6 glucose at the catalyst to biomass mass ratio of eight at 400 °C.*)

Action per 1-2: We apologies for overlooking the catalyst/biomass ratios in **Fig. 5**. Here again, all the results were consolidated to a fixed mass ratio of eight. The mass ratio has been added in the caption of **Fig. 5** on **Page 18**. In addition, on **Lines 303-304 Page 18: Fig. 2** we added "... *cellulose with different catalysts at a catalyst to biomass mass ratio of eight*". In **Section 3 Supplementary Results for Reaction Pathway, Page 14 of SI**, we also specified the mass ratios in the captions of **Fig. S1** and **Fig. S2** where the results for LGA are now presented.

Action per 1-3: Following the parametric screening results based on allose in **Fig. S6**, we have fixed the best catalyst as Pd-PdO/ZnSO₄ with a nominal Pd content of 1.1 mol%. For comparison, we also chose ZnSO₄ support only and the blank tests without the addition of catalysts. On **Lines 324-326, Page 19**, we have specified the catalyst and its mass ratio (eight).

Comment 2: For corncob and sugarcane bagasse: why Pd-PdO/ZnSO₄-0.4 mol% show better performance than Pd-PdO/ZnSO₄-1.1 mol%? (**Table 2**). As comparison, Pd-PdO/ZnSO₄-1.1 mol% is better than Pd-PdO/ZnSO₄-0.4 mol% for allose pyrolysis (**Fig S(6)**).

Response 2: First of all, here again, we have removed all the results for 0.4 mol% Pd for the two real biomasses to avoid the confusion, and more importantly, to narrow down the scope and content of this

paper, which otherwise would be too lengthy and chaotic, as mentioned by the reviewer before. In addition, considering the furfural yields from 0.4 mol% and 1.1 mol% Pd-doped ZnSO₄ are similar, 32.2 vs 32.9 wt% for corncob feedstock, and 18.3 vs 22.8 wt% for sugarcane bagasse in the original **Table 2**, and that the error of these data is around 3%, we believe it would be reasonable to only focus on the 1.1 mol% doping amount in this paper. As such, the results for 0.4 mol% Pd have been removed from **Table 2** on **Page 19**.

In terms of why the two natural biomasses require less Pd than the pure allose, we believe it is caused by a number of reasons that warrant a separate study. At this point, we believe the structure should be one of the influential factors. Compared to the pure allose, the two natural biomasses consist of both C5 and C6 sugars, and the C5 sugars can even readily decompose into furfural without the presence of Pd on ZnSO₄ (see xylan in **Fig. 4(c)**). Additionally, the presence of lignin could also be another reason, as the lignin char has proven catalytic for the decomposition of biomass (Chem. Eng. J 2022, 432: 134372). Again, a detailed study is essential to clarify the effect of feedstock.

Action: Results for 0.4 mol% Pd in **Table 2** were removed.

Comment 3: The proposed reaction pathways suggested that glucose should be the intermediate for the transformation of LGA to FF. Why FF yields from LGA are higher than that from glucose?

Response 3: Thank you for this invaluable question. Prior to answering this question, we would first summarise the difference between the two feedstocks, both dry and wet, with the use of the best catalyst (1.1 mol% Pd, mass ratio 8 and 400 °C). The yield of furfural was calculated based on **Eq 6** with the details presented in **Section 4.3** in the main manuscript.

	Glucose		LGA	
	Furfural yield, mol%	Furfural selectivity, %	Furfural yield, mol%	Furfural selectivity, %
Dry	77.3 (see Fig. 4(d))	60.6 (see Fig. 4(d))	39.5	31.1 (see Fig. S15(b))
Wet	Not conducted	60%	77.8	61.4 (see Fig. S15(b))

As can be seen, the results did support that the furfural yield and selectivity from glucose is higher than from the LGA. Upon the use of wet LGA, the furfural selectivity was also increased to a similar level with that from the dry glucose. These results support our proposed reaction pathways that glucose is an intermediate after the hydrolysis of LGA and even the real feedstock cellulose.

To address the reviewer's concern, following content has been added in the manuscript:

Lines 393-395, Page 23: Additionally, as the wet LGA has a similar furfural selectivity with that of the dry glucose at ~61% (see **Fig. 4(d)** and **Fig. S15(b)**), it is conclusive that **Step 4** should not be the rate control step for the overall conversion.

Comment 4: In 4.2 Catalyst synthesis, what is the propose of the annealing of $\text{ZnSO}_4 \cdot \text{H}_2\text{O}$ to anhydrous ZnSO_4 ?

Response 4: This is to remove the crystal water from $\text{ZnSO}_4 \cdot \text{H}_2\text{O}$ so as to create a anhydrous support that is identical with the Pd-laden ones. Otherwise, the effect of chemical water within ZnSO_4 could create an artifact during the comparison between different catalysts.

Comment 5: The catalyst was recycled by a tweeter, how about the recovery rate? The characterizations of recycled catalyst were not described.

Response 5: Firstly, approximately 80% of catalyst can be recovered from the Pyro-probe micro-reactor each time. However, for the large-scale fixed-bed reactor, we can recover almost all the catalysts, with a minimum recovery rate of 97%. Secondly, regarding the spent/recycled catalyst, we did observe it under SEM and TEM, but did not see any obvious changes. This can be partly reflected by its relatively stable activity in **Fig. S13**. More specifically, concerning the loss of S which otherwise would negate the stability of the catalyst, we have specifically analysed the recycled catalyst by XPS, with the results summarised in **Fig. S14**.

Action: On **Lines 295-300** in **Page 17**, we have updated the relevant descriptions as “*In addition, the S_{2p} XPS analysis results in **Figs. S14(a)** and **(a')** also confirmed an identical peak position and width between the fresh and spent Pd-PdO/ ZnSO_4 catalysts, proving the thermal stability of this sulphate-supported catalyst. In contrast, the ZnSO_4 support alone is rather unstable, displaying a broadened S_{2p} XPS spectrum with the formation of new species shown in **Figs. S14(b)** and **(b')**. Inferably, the unique structure in **Figs. 2-3** is accountable”.*

Reviewer #2 (Remarks to the Author):

This is an impressive work of Zhang and co-workers on the production of Furfural by means of Flash Pyrolysis starting from C6 Sugars promoted by the Pd-PdO/ ZnSO_4 Catalyst. The research area is of high relevance, being timely and a hot topic at the moment. In general, the paper is very solid: experiments are well conducted, a full phisico-chemical characterization of the catalyst was carried out and conclusions are consistent with experimental data. In my opinion, paper can be accepted almost as such even for a “big journal” such as Nature Communication.

A: Thank you very much for the invaluable comments and suggestions. We have revised the paper carefully and all the revisions are highlighted in **blue** in the main manuscript. Followings are the replies to the individual questions:

Some minor point:

Comment 1: Maybe the title is misleading. Maybe it is better to change into: “High Production of Furfural by Flash Pyrolysis of C6 Sugars and Lignocellulosic Biomasses promoted by Pd-PdO/ ZnSO_4 Catalyst”

Response 1: We appreciate the reviewer's suggestion. The title has been revised as suggested on the front page.

Action: Title was revised.

Comment 2: Some English polishing to correct typos and mistakes is needed.

Response 2: Thanks for the suggestion from the reviewer. The manuscript has now been carefully polished and proofread by every author. We had also sought paid editing service from the Research Square for the language polishing.

Action: English polishing was made by a paid service from the Research Square.

Otherwise, a very nice paper.

Reviewer #3 (Remarks to the Author):

The manuscript discusses the use of Pd supported on ZnSO₄ for converting carbohydrate pyrolysis vapors to furfural. Furfural is an important product molecule and the apparent ability of the Pd/ZnSO₄ to convert C6 carbohydrate-derived molecules to furfural is interesting.

There are, however, several significant concerns with the manuscript:

Comment 1: The quantification of furfural production is based on just considering its content in the measured product stream. **1-1)** There are no overall mass balances, so even though the portion of the product stream that is furfural increases with the use of the Pd/SO₄ catalyst, **1-2)** it is not clear if the overall product stream is reduced, i.e., what is the portion of the reactant that ends up in the GC measured product stream for each scenario (blank, ZnSO₄, Pd/ZnSO₄). **1-3)** Clearly, the yield of furfural as determined from just the product stream is significantly different than the yield of furfural based on the amount of reactant converted. This is a significant shortcoming of the manuscript.

Response 1: We apologize for the confusing and missing information in the original submission. Per your comments and those from other reviewers, we have revised the paper significantly. Regarding the three specific comments you raised here, our short responses are outlined below:

1-1) Overall mass balance – this has been now been added in as a separate **Section 5.4** in the Supporting Information (**SI**). The mass balances were provided for both Pyro-probe micro-reactor and fixed-bed reactor. In brief, for the Pyro-probe reactor, the mass balance of carbon (C) for the case of pure glucose (dry) under the conditions of using Pd-PdO/ZnSO₄ catalyst, 400 °C and catalyst to feedstock mass ratio of 8 was demonstrated. In this case, the products are relatively simple, mainly including furfural (80% yield), CO and CO₂ (~10% yield each) per mole glucose. The theoretical reaction for the conversion of glucose to furfural is also easy to establish. The description can be found in **Section 5.4.1** in the **SI**. Secondly, regarding the fixed-bed reactor, all the three products (gas,

solid and liquid) can be directly measured for the mass. Subsequently, the overall mass balance was easily determined by comparing these masses. Description can be found in **Section 5.4.2** in the **SI**.

Action: A separate **Section 5.4** was added into the **SI**.

1-2) Product stream entering into the GC column – All the experiments were conducted, both in Pyro-probe and fixed-bed reactor are batch-scale. The feedstock and catalyst were pre-loaded into the reactor, and the unconverted feedstock remained as solid char within the reactor. For the Pyro-probe micro-reactor, the liquid and gas products flow into the GC columns for real-time measurement, whereas for the fixed-bed reactor, the liquid product is further condensed in a cooling train, while the dry/permanent gas was measured by an on-line gas analyser. In this regard, the products entering the GC are always reduced in terms of its quantity. For the direct flow of the liquid into the GC columns in Pyro-probe, they are also assumed to fully (or nearly fully) flow into the column. This is because we are using a transferring line which is heated to 300 °C between the Pyro-probe reactor and GC, and the largest boiling point for the liquid product is only 260 °C for LGO (162 °C for furfural), which has been captured by our GC system (e.g. **Fig S7** in **SI**).

For the specific product amount entering into the GC columns for the three cases, the following **Table** summarizes the solid yield of glucose. The difference of it to 100 is indeed the amount including both gas and liquid flowing into the GC columns during the Pyro-probe testing. Additionally, we cross-checked these quantities with the results from the fixed-bed reactor where the liquid can be measured directly, which confirmed a good match between the two different set-ups.

Table Yield of solid from the pyrolysis glucose in the presence and absence of catalysts

	Catalyst to glucose mass ratio	Yield of solid, wt%	Error, wt%
Blank	-	32.5	0.9
ZnSO ₄	8	13.9	1.8
Pd-PdO/ZnSO ₄	8	9.3	1.8

Action: To improve the clarity on this point, the following revisions have been made:

Lines 201-202, Page 13: *Unless specified elsewhere, all the experiments were conducted in a batch scale with the feedstock and/or catalyst being pre-loaded into the reactor.*

Section 4.3 – Pyrolysis experiments in the manuscript, Pages 30-31: We specified the formation of solid and its yield calculation method.

1-3) Yield of furfural based on total liquid: We agree with that the yield based on total product stream is significantly different from the yield based on the reactant converted. Indeed, our yield of furfural was determined based on the quantity of feedstock, NOT based on the total product stream we produced. This aims to make our results comparable with the literature data which usually reports the yield of furfural based on the feedstock. More specifically, to determine the yield of furfural, we

first work out the total liquid yield (based on the total mass of feedstock), then multiply the total liquid yield and the concentration of furfural within it. Regarding the concentration of furfural within a liquid, it was determined by calibrating the area% to concentration based on the use of standard concentrations of furfural. To further validate the accuracy of this method, we also employed an external calibration method by injecting different quantity of furfural into the GC-MS to check its amount versus area. The results from both methods are very consistent.

Action: In **Section 4.3** of the revised manuscript (**Lines 511-542, Pages 30-31**), we have provided details for the calculation methods for product yields and selectivity for both Pyro-probe and fixed-bed reactors. Additionally, in the **Section 5.1 (Pages 18-22)** and **Section 5.4 (Pages 23-24)** of the **SI**, we have provided all the calibration details.

Comment 2: The definition of yield and selectivity is not at all clear in the main manuscript and requires digging into the supplementary information to find. As these definitions are seminal to understanding the reaction results, they must be clearly articulated in the main manuscript.

Response 2: We apologize for the confusion caused by having this important information in the **SI** of the original submission. Per your suggestion, we have moved the calculation methods for yields and selectivity into the **Section 4.3 (Lines 511-542, Pages 30-31)** in the revised manuscript. We also elaborated these methods in details in the manuscript. All the calibration curves still stay in the **Section 5.4 (Pages 23-24)** in the **SI**. To reiterate, the yields were calculated on the mass basis of feedstock, and the selectivity is on the area basis per the liquid spectra collected from GC-MS or FID. For instance, the furfural selectivity was determined by its area relative to the total areas for all the peaks in the spectra.

Action: Details of calculation method were added in the **Section 4.3 (Lines 511-542, Pages 30-31)** in the revised manuscript, and **Section 5.4 (Pages 23-24)** in the **SI**.

Comment 3: 3-1) Almost all of the reaction results are given for 1.1 mol% Pd/ZnSO₄ (Table 2 being the only exception). Additionally, extensive characterization is only performed on the 1.1 mol% material. Therefore, why is the introduction claiming doping levels from 0.4-1.1 mol%? Additionally, the 0.4 and 1.1 mol% materials gave similar results in **Table 2. 3-2)** Seems that the key point is Pd loading on ZnSO₄ leads to more furfural in the measured product and no statement is able to be made on what is a preferred Pd loading.

Response 3-1: We apologize for the massive yet messy/confusing information we have provided in the original submission. The primary rationale for us to test different amounts of Pd is that we wanted to optimise it. You are correct that the results for 0.4 mol% and 1.1 mol% are very much similar in **Table 2**. However, for some feedstocks, we did still observe slight variation of the optimum Pd amount. As this causes a lot of confusion, we have decided to remove all the results for the 0.4 mol% including those in **Table 2** in the revised paper. The only exception is the screening test results we summarised in **Fig. S6 (Page 7)** in the **SI**. Regarding the reason for the varying optimum amount of Pd for different feedstocks, we will further examine it in detail in our future work.

Action:

Lines 25-26, Page 2: Herein, we report a novel and reusable heterogeneous catalyst, Pd-PdO/ZnSO₄ for the doping of 1.1 mol% palladium on ZnSO₄...

Lines 70-71, Page 4: Herein, we report a novel heterogeneous catalyst, namely Pd-PdO/ZnSO₄ with the doping of around 1.1 mol% palladium (Pd) on zinc sulphate (ZnSO₄) ...

Lines 453-455, Page 28: A novel, bifunctional and reusable Pd-PdO/ZnSO₄ catalyst with an optimum Pd loading of 1.1 mol% was designed...

Response 3-2: We apologize for making this unclear in the original submission. Previously, we had included the screening test results including the optimum Pd loading amount in the **Fig. S6** of **SI**. We believe this is the reason it was missed out during your review. To make this clearer, we have added detailed discussion as a separate paragraph for the screening test results in **Section 2.2, Page 13** in the manuscript.

Action: A separate paragraph detailing the screening test results including optimum Pd loading was made in **Section 2.2 (Lines 198-212, Page 13)** in the main manuscript. The respective results can be found in **Fig. S6, Page 7** in the **SI**.

Comment 4: Lines 245-248 discuss the possible role of water in breaking the glycosidic bonds for the cellulose pyrolysis step. Why is the Zn²⁺ result being discussed? In the pyroprobe experimental setup the Pd/ZnSO₄ catalyst is only in contact with the pyrolysis vapors.

Response 4: Thank you for the valuable question, and we agree with you that the role of Zn²⁺ here is irrelevant. As such, we have deleted the relevant discussion and **Fig. S11**. Alternatively, we discussed hydrolysis of wet cellulose by citing a new reference [51] and conducting the TGA-MS for dry and wet cellulose. The TGA-MS results were added as **Fig. S10(e)** in the **SI**, which confirmed an easier decomposition of the wet cellulose for a quicker release of all the products and even an inhibition of the formation of LGO. All these results further consolidated the importance of hydrolysis and our proposed reaction pathway.

Action: The relevant content in **Lines 245-248, Page 16** in the **original submission** has been deleted accordingly. **Fig. S11** in the **SI** for the proposed mechanism for the ZnCl₂ has been deleted accordingly as well. Alternatively, extra TGA-MS data for the dry and wet cellulose were plotted as **Fig. S10, Page 10** in the **SI**.

REVIEWERS' COMMENTS

Reviewer #1 (Remarks to the Author):

The authors have made fruitful progress in the revised manuscript, which enabled the manuscript more logical and readable. After careful consideration, I agree, in principle, to accept it by Nature Communications. The authors should double check it before next submission.

1 Both Figures 2 and 3 contain TEM images, can they be combined?

2 In some chemdraw pictures, some atoms overlapped and some bonds are distorted, which do not reach the requirement of an important journal.

3 There are plenty of syntax problems in the manuscript.

Reviewer #3 (Remarks to the Author):

The authors have adequately addressed the comments from the reviewers.

Reviewer #1 (Remarks to the Author):

The authors have made fruitful progress in the revised manuscript, which enabled the manuscript more logical and readable. After careful consideration, I agree, in principle, to accept it by Nature Communications. The authors should double check it before next submission.

Answer: We greatly appreciate the reviewer's invaluable comments. Per the further comments received, we have revised the paper with all the revisions track changed in the main manuscript. The individual replies are provided point by point below.

Comment 1: S Both Figures 2 and 3 contain TEM images, can they be combined?

Reply 1: Thank you for your kind suggestion. Per your suggestion, we have considered merging the TEM images in the two figures together. However, we found this would instead misalign with the information we aim to convey through each figure. **Fig. 2** aims to provide the information for the speciation of Pd and S within the fresh catalyst, whereas **Fig. 3** for TEM mapping has a different purpose for the overall distribution of the individual elements across several particles. Therefore, instead of merging all the TEM images together, keeping them differently provides a more consistent and clearer flow of the paper.

Comment 2: In some chemdraw pictures, some atoms overlapped and some bonds are distorted, which do not reach the requirement of an important journal.

Reply 2: We apologies for the mistake. All the ChemDraw files in **Figs. 1, Fig. 6** and 7 have now been checked carefully and revised accordingly in the manuscript.

Comment 3: There are plenty of syntax problems in the manuscript.

Reply 3: Thank you for pointing it out to help us further improve the clarify of our paper. We have sought paid AJE digital editing service offered by the Research Square Company. All the changes can be found in the tracked manuscript and supporting information document. A careful proofreading has been further made thoroughly by us prior to the final submission.

Reviewer #3 (Remarks to the Author):

The authors have adequately addressed the comments from the reviewers.

Answer: Thank you very much. We sincerely appreciate your invaluable comments which have improved the quality of our paper greatly.